# Subtypes and Mechanistic Advances of Extracorporeal Membrane Oxygenation-Related Acute Brain Injury

**DOI:** 10.3390/brainsci13081165

**Published:** 2023-08-04

**Authors:** Bixin Deng, Junjie Ying, Dezhi Mu

**Affiliations:** 1Department of Pediatrics, West China Second University Hospital, Sichuan University, Chengdu 610041, China; dengbixin@stu.scu.edu.cn; 2Key Laboratory of Birth Defects and Related Diseases of Women and Children, Sichuan University, Ministry of Education, Chengdu 610041, China; yingjunjie177@163.com

**Keywords:** extracorporeal membrane oxygenation, critical illness, acute brain injury, morbidity, hemodynamics

## Abstract

Extracorporeal membrane oxygenation (ECMO) is a frequently used mechanical cardiopulmonary support for rescuing critically ill patients for whom conventional medical therapies have failed. However, ECMO is associated with several complications, such as acute kidney injury, hemorrhage, thromboembolism, and acute brain injury (ABI). Among these, ABI, particularly intracranial hemorrhage (ICH) and infarction, is recognized as the primary cause of mortality during ECMO support. Furthermore, survivors often suffer significant long-term morbidities, including neurocognitive impairments, motor disturbances, and behavioral problems. This review provides a comprehensive overview of the different subtypes of ECMO-related ABI and the updated advance mechanisms, which could be helpful for the early diagnosis and potential neuromonitoring of ECMO-related ABI.

## 1. Introduction

Extracorporeal membrane oxygenation (ECMO), a life-saving modality first proposed in 1960s [1], has been widely used in neonates to adults, especially those with severe disease who are unresponsive to conventional therapies or those requiring organ transplantation. The cannulation method divides ECMO into two modes: venovenous (VV) and venoarterial (VA)-ECMO. In VV-ECMO, blood is drained from a large vein, circulated through a centrifugal pump and semi-permeable membrane for gas exchange, and returned via a vein, supporting pulmonary function in patients with refractory hypoxemia. VA-ECMO is similar, including a centrifugal pump and oxygenator; however, oxygenated blood is returned via an artery, providing heart and lung support for patients lacking regular heart functions [2]. Central VA-ECMO canulation was conducted via direct venous penetration into the right atrium or direct arterial cannulation into the ascending aorta. On the other hand, peripheral VA-ECMO, referring to femoro-femoral cannulation, femoro-axillary cannulation, jugular-axillary cannulation, and jugular-carotid cannulation [3].

Despite the potential benefits of ECMO, this life-saving technology may be associated with serios complications, including acute brain injury (ABI) [4]. Acute brain injury encompasses several clinical neurologic events, including intracranial hemorrhage (ICH), ischemic stroke, hypoxic–ischemic brain injury (HIBI), cerebral edema, seizure, and brain death. Studies based on the Extracorporeal Life Support Organization Registry (ELSO) have reported ABI in 11–20% of adult patients undergoing VA-ECMO [5,6], and the most recent meta-analysis excluding data from ELSO up to 12.5% [7]. However, the actual incidence may be higher due to a lack of standardized neuromonitoring, limited use of computed tomography (CT) and/or magnetic resonance imaging (MRI) during ECMO, and challenges in the neuroimaging of patients supported by ECMO [8,9]. Previous research indicates that patients undergoing VA-ECMO for cardiac support have a higher ABI rate than those undergoing VV-ECMO for respiratory failure [10,11], with neonates [12] and preterm infants [13] being particularly susceptible. Notably, autopsy studies have reported a significantly higher occurrence of ECMO-related ABI, ranging from 68% to 81% [2,14,15].

Cerebral injuries significantly impact hospital mortality, with rates up to 95% and 100% in patients with two or three or more central nervous system (CNS) complications, respectively [5]. According to a study using the international ECMO registry data, the survival rates for adult VA-ECMO-treated patients with radiographically confirmed infarction or hemorrhage were 17.4% and 10.5%, respectively, compared to 57% in those without cerebral lesions [16]. Survivors also have a relatively high probability of subsequent disability [4]. This review provides an overview of the patterns, possible mechanisms, and monitoring methods of ECMO-induced ABI.

## 2. Patterns of ECMO-Related ABI

The characteristics of each subtypes of cerebral injuries in patients with ECMO are not well described. In studies reporting brain autopsy data, ECMO-related ABI was found in 68% of adult patients and 79% of infants, with HIBI reported as the most common ABI in both populations, followed by intracranial hemorrhage and ischemic infarction [14,15].

### 2.1. Intracranial Hemorrhage

ICH (including micro- and macro-hemorrhages) was diagnosed in 22/43 (51%) of decedents during autopsy [2], while, in retrospective studies, the prevalence of ICH was similar between VV- and VA-ECMO patients (2–18% vs. 4–19%, respectively) [5,10,17,18,19,20]. Pathologically, ICH occurs more frequently in the frontal neocortex, basal ganglia, cerebellum, and pons [2]. As reported, the most common type of intracranial hemorrhage is subarachnoid hemorrhage (SAH), followed by petechial intraparenchymal hemorrhage [14,19]. However, the ICH timing is not yet well characterized. In one study, 85% of ICH presented on a CT scan shortly after cannulation [19], while, in another study, the median time for ICH detection was 7 days after ECMO initiation (see Table 1) [18].

Moreover, the mechanisms underlying ICH are not well understood. A previous study reported that two steps are involved. First, the integrity of endothelial cells and the blood–brain barrier is altered, following the extravasation of plasma and erythrocytes, impairing hemostasis [21]. In addition, bloodstream infection is correlated with ICH in patients using a left ventricular assistive device; therefore, it may also play a role in ECMO patients [22,23]. Furthermore, a previous study found that, although the hemorrhagic group had the highest average blood pressure, it had the lowest cerebral tissue oxygenation saturation, suggesting that elevated vascular resistance may lead to poor cerebral perfusion [24]. The risk factors related to ICH include use of anticoagulants and antiplatelet therapy [22], female sex [25], higher pre-cannulation PaCO2 [26], thrombocytopenia, high transfusion requirements [18], a large dual-lumen venovenous cannula [20], ECMO duration [17], bloodstream infections [22], renal failure, and dialysis [27]. Finally, ICH patients are known to have high mortality and morbidity [18].

### 2.2. Ischemic Stroke

In a mixed VA- and VV-ECMO study, ischemic stroke was observed in 4.1% of patients across all age groups [28]. Although ischemic stroke is more common in VA ECMO (3.6–5.3%), it also occurs in VV ECMO patients (1.7–2%) [5,10,25,27,28]. However, one study reported that the true prevalence is also underestimated, and a rate of 11% was reported recently [29]. The pathology of ischemic stroke, occurring in the frontal neocortex, basal ganglia, anterior hypophysis, and cerebellum [2], is territorial, such that embolism may be the underlying mechanism [30,31]. Moreover, one study claimed that ischemic stroke events during ECMO are located in the left hemisphere in 70% of cases [32].

Although poorly described, cerebral venous sinus thrombosis and emboli from the circuit through the patent foramen ovale are possible etiologies of ischemic stroke [33]. In addition, researchers reported that it tends to occur 1 week after ECMO cannulation, with the only risk factor being platelets >350 giga/L at ECMO initiation [25]. However, other studies have concluded that multiple factors likely contribute to ischemic stroke, including a low brain perfusion state, a dual-lumen venovenous catheter [25,34], hemolysis [35], internal jugular vein cannulation [33], and acute infection or sepsis [36,37], which can lead to thromboembolization. Additionally, in an animal study, hyperthermia was shown to be associated with ischemic cerebral injury in the hemisphere ipsilateral to ligation, leading to elevated cerebral metabolism [38].

### 2.3. Seizures

Seizures are abnormal, paroxysmal electroencephalography (EEG) events that vary from background activity [39]. Due to the use of sedative drugs and a lack of EEG monitoring, seizure incidence may also be underestimated at 2–6% in patients undergoing VA ECMO [5,26] and 1.3% in VV ECMO adult patients [40]. Previous studies have claimed that younger pediatric patients have higher occurrence of seizures than older patients [41,42]. Other studies using continuous EEG and regular sedation reported a much higher seizure rate in patients [43,44,45]. Seizure activity can be subsequently detected in patients with ICH, embolic infarcts, or global cerebral hypoxia/anoxia [46]. However, most ECMO-related seizures are related to neurologic insults, occurring around the time of cannulation [45]. It is noteworthy that, in children and adults under ECMO, severe background abnormalities (burst suppressions, severe slowing or unresponsiveness, or electrographic seizures) result in CT- or MRI-defined brain injury and poor outcomes, including lower IQ and neurodevelopmental delays [47,48,49,50]. Importantly, severely abnormal background activity in the first 24 h after ECMO initiation is associated with death and seizure lateralization related to arterial cannulation position, including the middle cerebral artery (MCA) territory. However, in patients presented with seizures, mostly implicating brain injury, only one-third of patients with ABI had monitored seizures [39]. Moreover, cerebral gas embolism (CGE) may result in the rapid onset of epileptic seizures [51], and cerebral edema is another known risk factor for seizures [52].

### 2.4. Hypoxic–Ischemic Brain Injury (HIBI)

As the most common type of ABI in patients who received ECMO at autopsy [14,15], HIBI is reported to have a higher incidence in patients with VA-ECMO over VV-ECMO (13% vs. 1%, *p* < 0.001) [53]. Pathologically, HIBI demonstrates the most diffuse damage in multiple anatomic locations [2], involving the cortex, cerebellum, brain stem, and basal ganglia, with a preponderance in the cerebral cortices (82%) and cerebellum (55%) [14]. Hypoxia from refractory respiratory failure is a possible cause, while CO_2_ dysregulation, cerebral vasoconstriction, and North–South syndrome, resulting from hemodynamic changes specific to VA-ECMO, can lead to cerebral ischemia [33]. When it comes to risk factors, HIBI is most likely to occur in patients with hypertension, hyperlactatemia, and low pH (acidosis), and signs of inadequate perfusion. It often presents during the peri-ECMO cannulation period (24 h) before and after ECMO cannulation [14], because ECMO provides adequate perfusion of the brain after successful cannulation.

### 2.5. Brain Death

The occurrence of brain death is reportedly different in the two modes of ECMO. Approximately 2% in patients with VV-ECMO and 7.9–13.1% in patients with VA-ECMO [5,7,54]. Contrary to the prevalence of ABI in different populations, the incidence of brain death in pediatric patients increases with age, from 1.3% in those aged 0–30 days, to 4.0% in those aged 31 days to less than 1 year, and finally to 8.4% in those aged 1 year to less than 16 years [55]. Additionally, severe post-anoxic swelling, extensive infarction, or hemorrhage may lead to brain death [5,36]. Traditionally, brain death was defined as irreversible functional loss of the entire brain, including the brainstem [56]. Clinical assessments of brain death include hemodynamic stability (systolic arterial pressure >90 mmHg), adequate core temperature (>36 °C), and without circulating analgesics, sedatives, muscle blocking agents, or severe electrolyte or glucose disturbances [57]. Unfortunately, in the presence of ECMO support, brain death determination is challenging and there are no guidelines for the determination of brain death in ECMO [46]. Moreover, novel monitoring methods such as EEG, plasma biomarker glial fibrillary acidic protein (GFAP), and cerebral angiography may be useful; however, they still require further investigation [58,59,60].

## 3. Possible Mechanisms of ABI in ECMO

### 3.1. Common Mechanisms

#### 3.1.1. Altered Hemostasis

As blood components come into contact with the extracorporeal surface, they inevitably activate the coagulation system (Figure 1), which may lead to endothelial injury and disrupted microcirculation; thus, anticoagulation and antiplatelet drugs are often used in this condition [61]. The intrinsic coagulation pathway is initiated by factor XII, combined with factor XI, high-molecular-weight kininogen (HMWK), and pre-kallikrein, which are components of the contact system. Factor XII is cleaved into factor XIIa and factor XIIf. Factor XIIa, achieving maximal levels within 10 min of ECMO initiation, not only triggers the following steps in the coagulation cascade but also converts HMWK and pre-kallikrein into bradykinin and kallikrein, respectively. Both promote XII and XI activations [62,63]. On the other hand, the extrinsic pathway is thought to be less important for this process. However, regardless of the ex- or intrinsic pathway, they share one activator, FXa, which converts prothrombin to thrombin to form clots [64].

Unfractionated heparin (UFH), the gold standard and frequently used anticoagulant during ECMO, downregulates thrombin and activated factor X (factor Xa) by enhancing the antithrombin activity. Meanwhile, activated partial thromboplastin time (APTT), activated clotting time (ACT), anti-factor Xa activity levels (anti-Xa), blood drug concentration, and viscoelastic tests are used as anticoagulation monitoring methods [65]. Of these, a range of 1.5 to 2.5 times the patient’s baseline APTT (40–80 s) or a goal of 180–220 s for ACT is mostly used in cases of thromboembolism or bleeding risk [66,67]. However, the most appropriate anticoagulation monitoring tests is still missing, and recently, it was reported that the levels of APTT did not show any association with bleeding or thrombosis [29,68]. In addition, low-molecular-weight heparin (LMWH) is widely used to prevent and treat venous thromboembolism. Although it targets factor Xa, it also activates antithrombin, exerting an anticoagulant effect [65].

Platelets promote hemostasis and accelerate clotting. It is noteworthy that, in one study, the platelet count dropped to 150 g/L on day 5 from 221 g/L prior to ECMO initiation [69], A previous study reported that platelet loss during ECLS is mainly caused by platelet activation and blood draw for diagnostic phlebotomy, while approximately one-third of the cause of platelet loss remains unknown [70]. Moreover, platelets have been shown to be dysfunctional in ECMO-supported critically ill neonates despite platelet transfusion [71].

Because the ECMO circuit provides a large surface of area of blood interface and its composition, thrombi from it are noteworthy. A previous study showed that ECMO-associated venous thromboembolism rates ranging from 18 to 85% in different centers [72]. These thrombi were seen in oxygenator membrane surface, arterial tubing venous tubing, and connector thrombi [73], and when the thrombi returned to the body they caused damage to multiple organs.

However, inadequate systemic anticoagulation increases the risk of bleeding in cases of overdosing or thrombosis in case of underdosing. Therefore, it is essential to monitor anticoagulation and balance procoagulant and anticoagulant factors [74,75]. Viscoelastic hemostatic assays (VHA) and whole blood point-of-care coagulation assays are used to measure the viscoelastic properties of the clot. Of these, thromboelastography (TEG) and thromboelastometry (ROTEM) are the only two VHAs validated for clinical use, both are equipped with multiple assays, allowing extrinsic and intrinsic clot activation, the assessment of the contribution of fibrinogen formation and heparinase assays. Although VHA has only been validated as a guideline for therapeutic administrations, the hypercoagulable state demonstrated by TEG and ROTEM has also predicted a high risk of thrombotic complications in several studies [76,77]. In addition to underlying disease and anticoagulant therapy, different configurations of ECMO may also alter hemostasis. Compared to VV-ECMO, patients on VA-ECMO have significantly decreased platelet counts, fibrinogen, antithrombin, and clot strength with higher D-dimer levels, which are in parallel with consumptive coagulopathy. The rate of thrombosis is approximately 40% in VA-ECMO, while in VV-ECMO, it is doubled [78].

#### 3.1.2. Immune System Activation

Although patients receiving ECMO support have severe disorders that can lead to immune system activation, exposure to the artificial surface of the ECMO circuit and disrupted cerebral autoregulation can also alter the immune system, inducing the expression of various immune cells and cytokines [79]. Consequently, systemic inflammatory response syndrome (SIRS) can be induced following a series of fatal complications, including bleeding, thrombosis, and capillary leakage syndrome [64].

The components of the contact system also drive inflammation and directly activate neutrophils [62]. Moreover, thrombin, a vital product of the coagulation system activated by the contact system during ECMO, is the main activator of platelets, which secrete several soluble mediators and bind to leukocytes [64]. The complement system, at the forefront of the innate immune response, is activated following ECMO initiation. Moreover, among the three activation pathways (the classical pathway (CP), alternative pathway (AP), and lectin pathway (LP)), the AP has been shown to be the prominent mechanism of complement activation [80], responsible for the production of proinflammatory anaphylatoxins C3a, C5a, and the membrane attack complex. This, in turn, leads to increased vascular permeability, smooth muscle contraction, leukocyte recruitment, and the production/release of other inflammatory cytokines [81].

Although phagocytes are also activated during ECMO, they do not accumulate in the extracorporeal circuits or release humoral factors [82]. In a porcine ECMO model of induced SIRS, mast cells in the intestinal mucosa are thought to be responsible for a rapid increase in TNF-α and IL-8 immediately after ECMO initiation, as evidenced by higher sensitivity in detecting inflammation during ECMO compared to blood leukocyte and C-reactive protein counts [83].

In addition, neutrophils are also activated in ECMO, with the activators shown to be kallikrein [62], Xa, TNF-α, thrombin [64], and C5a [84], resulting in the attracting and priming of neutrophils and adherence to the endothelium [84]. Widespread microvascular injury and multiorgan dysfunction are possible by adhering to the capillary/venular endothelium and producing cytokines, arachidonic acid metabolites, and reactive oxygen species [85,86]. Another study, using ex vivo models of ECMO, showed that neutrophils are activated within the first 30 min of ECMO initiation; however, leukocytes do not adhere to the extracorporeal circuit [82]. Although the autoreactivity of B and cytotoxic T cells to CNS-specific antigens was detected, parallel to ECMO treatment [79], their function in ECMO-related brain damage remains unknown.

Elevated IL-8 can act as a powerful trigger to induce the adhesion of monocytes to the vascular endothelium [87]. Another cytokine, IL-6, which acts as both a pro- and anti-inflammatory cytokine, is also elevated in adult and pediatric ECMO patients [64]. However, studies have suggested that increased IL-6 levels are associated with lung injury [88] and poor survival rate [89].

Although activated neutrophils levels are significantly increased, the high number of immature neutrophils responsible for immunosuppression is higher in ECMO patients, resulting in a high risk of infection. In addition, both mature and immature neutrophils express low levels of the C5a receptor [90], which is related to their ability to kill gram-positive cocci [91]. In addition to abnormal neutrophils, monocyte dysfunction, T-cell dysfunction, and the expansion of myeloid-derived suppressor cells (MDSC) are also observed under these conditions, with powerful immunosuppression [92,93] or immunoparalysis. Notably, a recent study suggests that it may be associated with worse outcomes in patients receiving ECMO (Figure 2) [94].

### 3.2. Mechanism of ABI Specific to VA-ECMO

#### 3.2.1. Hemodynamics in ECMO

There are multiple choices of cannulation artery, with the right common carotid artery being the most commonly used in infants [95]. While peripheral ECMO is more common in the elderly, the blood flow from the oxygenator to the vessels is not in accordance with intrinsic cardiac flow, such that the blood flow from the two points of circulation will meet. This “mixing cloud” is determined by the amount of ECMO support provided and the degree of left ventricular ejection (Figure 3) [96]. Thus, in the presence of pulmonary dysfunction, the blood ejected from the left ventricle is not well oxygenated. This differential [97] can lead to hypoxia in organs perfused by the proximal aortic branches, including the coronary and innominate arteries. For example, if the mixing cloud is located between the brachiocephalic trunk and the left common carotid artery, then the two cerebral hemispheres will be perfused by blood with different levels of oxygen, velocity, pressure, and pulsatility [98,99].

#### 3.2.2. Cannulation Method

In adult, central ECMO (blood returned to the body via an ascending aortic cannula) is associated with left hemisphere lesions, while peripheral ECMO (blood returned to the body via a right common carotid artery cannula pointing toward the arch of the aorta) is related to right hemisphere lesions; predominantly located in the MCA territory [39]. Other researchers have found significantly increased diastolic and mean cerebral blood flow (CBF) velocities in the internal carotid artery contralateral to the cannulation side compared with the pre-ECMO level and in the control group [100]. A similar result is seen in animal experiments, where CBF [38] showed a significant decrease in the hemisphere ipsilateral to the cannulation side. One study [101] reported that neurologic injury in central ECMO patients, including ICH, increased intracranial pressure, and ischemic injury, is detectable just after higher CBF velocity is monitored. In addition, two other asymmetrical distributions of blood flow, significantly higher blood flow velocity in the right rather than left posterior cerebral artery (PCA), and higher systolic flow in the left MCA after ECMO cannulation rather than in the right MCA are reported in support of the fact that hemorrhagic and ischemic injuries tend to, respectively, occur in the left and right hemispheres [95].

#### 3.2.3. Changes in Cerebral Autoregulation

A healthy brain has the ability, known as cerebral autoregulation, to maintain a relatively stable blood flow [102], which may be measured through the cerebral oximetry index (COx). A changing correlation between artery blood pressure (ABP) and cerebral oximetry approaches +1 when pressure autoregulation is lost, and cerebral tissue oxygen saturation becomes passive to changes in ABP. In the presence of intact pressure autoregulation, tissue oxygen saturation is no longer passive to ABP, and COx fluctuates around zero [103]. The impairment of cerebral autoregulation during ECMO has been shown to be correlated with neuroimaging abnormalities [104]. Studies measuring COx in ECMO patients found that ischemic patients had elevated COx values and low ABP associated with impaired autoregulation [24,105]. However, cerebral autoregulation can be altered by several factors such as acid-base status, blood carbon dioxide levels, oxygen saturation, pulsatile flow patterns, and blood pressure [24], some of which may occur before ECMO cannulation.

#### 3.2.4. Changes in Pulse Pressure

Pulse pressure, a dynamic sign of cardiovascular function, is often disturbed in patients receiving VA-ECMO. Specifically, low pulse pressure, defined as <20 mmHg throughout the first 12 h after ECMO initiation, is associated with ABI in patients supported by VA-ECMO [106]. The blood flow in patients supported by VA-ECMO is non-pulsatile, which is inferior to pulsatile perfusion in preserving microcirculation, resulting in collapsed capillaries, activation of inflammation, elevated lactate, and microvascular shunting from decreased hemodynamic energy [107,108]. In addition, increased leukocyte adherence and reduced endothelial-derived nitric oxide from decreased/irregular shear forces [107,109] have been observed. Reduced nitric oxide results in vasoconstriction, neutrophil activation, and expression of cellular adhesion molecules [107]. Other studies have indicated that reduced pulsatility correlates with endothelial cell dysfunction and integrality, presented by increased tight junction biomarkers ZO-1 and Occludin [110], resulting in capillary leakage and vascular permeability breakdown [111]. This kind of dysfunction can be reversed by the restoration of pulsatility [112].

### 3.3. Mechanisms of ABI Specific to VV-ECMO

There are different mechanisms in the pathophysiology of brain injury between patients receiving VV-ECMO and VA-ECMO. Due to different cannulation methods, the hemodynamic changes are less pronounced in VV-ECMO than in VA-ECMO. Furthermore, a previous study has confirmed that it is the blood flow, not the cannula position, that has a major impact on VV-ECMO efficacy [113]. Moreover, abrupt O_2_ and CO_2_ changes at ECMO initiation is specific to VV-ECMO. Because of the ability of cerebral blood flow regulation that CO_2_ possesses, a sudden change in PaCO_2_ at ECMO onset would influence the blood flow, which could precipitate brain damage [114], as the ICH and HIBI occurred shortly after ECMO cannulation or during the peri-cannulation period.

### 3.4. Damage Caused by Underlying Disease

Patients who require ECMO support suffer from critical illness, such as cardiac shock, cardiac arrest, or respiratory failure. Consequently, there is no doubt that these severe disorders can also lead to brain damage. Thus, cerebral lesions diagnosed with during or after ECMO support could precede ECMO, with the ECMO circulation aggravating brain damage.

For example, hypoxia can also be caused by underlying diseases. Normally, almost a third of cardiac output is received by the central nervous system, which means the brain is sensitive to anoxia. As one of the major indications for VA-ECMO, CA itself can also cause brain injury through following steps: a “no-flow period”, which represents the stop of cerebral blood flow right after the onset of CA; a “partial-flow period”, representing the phase when the cerebral blood returns to 25–40% of baseline; and a “reperfusion injury”, resulting from the return of spontaneous circulation [115]. In respiratory failure or ARDS, patients also experienced hypoxia, hypercapnia, and respiratory acidosis, which are involved in the onset of brain injury. Moreover, in patients with ARDS, right ventricular dysfunction occurred in over 20% patients [116]. The cardiac output was severely affected, which caused hemodynamic compromise, resulting systemic malperfusion and multi-organ failure [117].

Apart from hypoxia, other conditions such as inflammation, infection, acidosis, and electrolyte disturbances, are also involved in this process [27].

### 3.5. Neuromonitoring Based on the Possible Mechanisms

In the presence of ECMO support, traditional neuromonitoring methods are limited. Patients on ECMO with intracranial lesions may not exhibit neurological symptoms prior to diagnosis [118,119]. Moreover, patients with normal CT neuroimaging exhibit a 67% rate of abnormalities [2]. Therefore, it is critical to implement accurate neuromonitoring methods. However, understanding the mechanisms of ECMO-related ABI would be helpful for the early detection and intervention of ABI, which can improve the prognosis.

Several neuromonitoring methods has been tested in studies. For example, transcranial doppler ultrasound (TCD) can detect micro embolic signals (MES) in intracranial arteries [120] and monitor CBF velocity [101]. Continuous electroencephalography (cEEG) continuously monitors brain activity [39]. Near-infrared spectroscopy (NIRS) can be used to monitor brain perfusion, regional cerebral oxygen saturation (rScO2), and hemodynamic stability [121]. Furthermore, brain injury biomarkers, S100B [118], neuron-specific enolase (NSE) [122], GFAP [59], and tau protein [123] are proven to have an elevated level in ECMO patients with intracranial lesions. Notably, the diagnostic accuracy of combined monitoring modalities, such as EEG plus neuroimaging, plasma brain injury biomarkers plus Doppler ultrasound, and cerebral oximetry plus Doppler ultrasound, is higher than that of cEEG, NIRS, TCD, or plasma biomarkers alone [124].

## 4. Future Perspectives and Development

### 4.1. Anticoagulants Monitoring

Patients receiving ECMO with bleeding or thrombotic complications are associated with increased morbidity and mortality [125,126]; thus, adequate anticoagulation strategies and monitoring methods are crucial in avoiding these complications. Individualized anticoagulation therapy, the development of extracorporeal circuit materials, and multicenter studies may improve the effectiveness of anticoagulation therapy. Additionally, due to the shortages (which can be affected by other factors, such as high variability, etc.) of APTT and ACT in the monitoring of anticoagulant, novel methods, such as an anti-Xa assay, and the VHA has been used in many clinical laboratories [76]. For example, it is reported that the application of the anti-Xa assay for heparin monitoring in cardiac indicated, neonate [127], and pediatric [128] patients receiving ECMO was associated with a decreased bleeding rate, while another study has reported that patients in the TEG group tended to have fewer bleed complications; moreover, the heparin dosing was lower in the TEG group, and thrombotic complications did not increase compare to the APTT group [129].

### 4.2. Therapeutic Hypothermia

Therapeutic hypothermia is proven to be associated with improved survival and better neurological outcomes in neonates with moderate-to-severe hypoxic–ischemic encephalopathy [130], and it has been used in ECMO patients to reduce cerebral complications. Researchers claimed that early mild hypothermia exerted a positive effect on neurological outcomes in both children and adult ECMO patients [131,132]. On the other hand, mild hypothermia in neonates treated by ECMO did not improve the outcomes. This opposing outcome may be due to different populations and studies [133].

### 4.3. Hemoadsorption in ECMO

In the context of critical illness and ECMO support, patients often develop sepsis and a state of hyperinflammation, which are thought to be associated with poor prognosis, including brain damage [134,135]. Therefore, clinical interventions targeting inflammation are required. Hemoadsorption, which can remove cytokines, bilirubin, myoglobin, and certain medications, seems a promising novel therapeutic principle [136]. And it has been confirmed that VA-ECMO-integrated hemoadsorption treatment is associated with accelerated recovery from multiorgan and microcirculatory dysfunction, reduced inflammatory response, fewer hemorrhagic complications, and a lower risk of early mortality [137].

## 5. Conclusions

In conclusion, this study reviewed the characteristics of different subtypes and possible underlying mechanisms of ECMO-related ABI, including VA- and VV-ECMO, and VA- or VV-ECMO-specific mechanisms. The early detection of ECMO-related ABI is possible using combined neuromonitoring methods.

## Figures and Tables

**Figure 1 brainsci-13-01165-f001:**
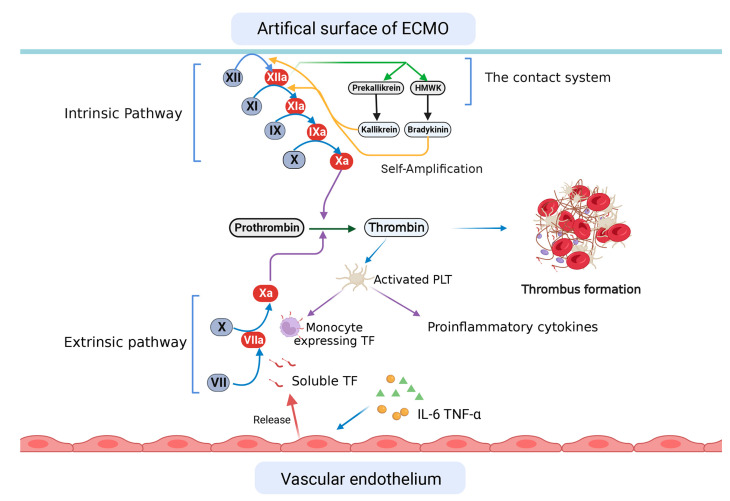
Coagulation activation in patients supported by extracorporeal membrane oxygenation (ECMO). When blood comes into contact with the artificial surface of the ECMO system, the intrinsic coagulation pathway is activated. The products of this contact system, kallikrein and bradykinin, exert a self-amplification function. Additionally, the extrinsic pathway, starting with the tissue factor released from endothelial cells and expressed on monocytes, plays a role in thrombus formation. HMWK: high-molecular-weight kininogen; TF: tissue factor; PLT: platelet.

**Figure 2 brainsci-13-01165-f002:**
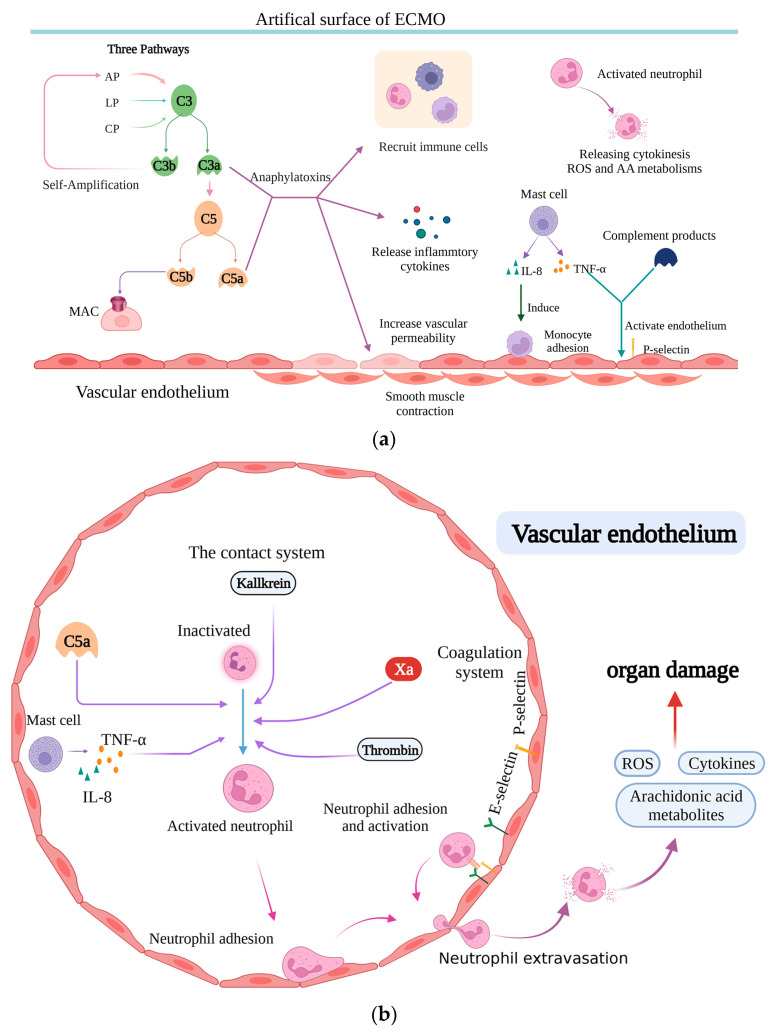
(**a**) Immune activation in patients supported by extracorporeal membrane oxygenation (ECMO). In addition to the underlying disorders, contact between the blood and the biomaterial of ECMO inevitably alters the immune system. The complement system is first activated; the alternative pathway (AP) is the main activation pathway, and C3 and C5 are cleaved into C3a, C3b, C5a, and C5b. Both C3a and C5a are anaphylatoxins that can increase leukocyte recruitment, vascular permeability, smooth muscle contraction, and the release of inflammatory mediators. Moreover, various immune cells, such as monocytes, mast cells, phagocytes, and neutrophils, are involved and exert vital effects. (**b**) The interaction between immune and coagulation systems in patients supported by ECMO. Neutrophil, activated by several factors, such as FXa, thrombin in the coagulation system, kallikrein in the contact system, and C5a and TNF-α in the immune system, is thought to play a vital role in organ damage by releasing stores of cytotoxic enzymes, cytokines, arachidonic acid metabolites, and reactive oxygen species. Additionally, when activated by thrombin, PLT conjugates with leukocytes, especially neutrophils, and monocytes. Thus, there is an interaction between the immune and coagulation systems. LP: lectin pathway; CP: classical pathway.

**Figure 3 brainsci-13-01165-f003:**
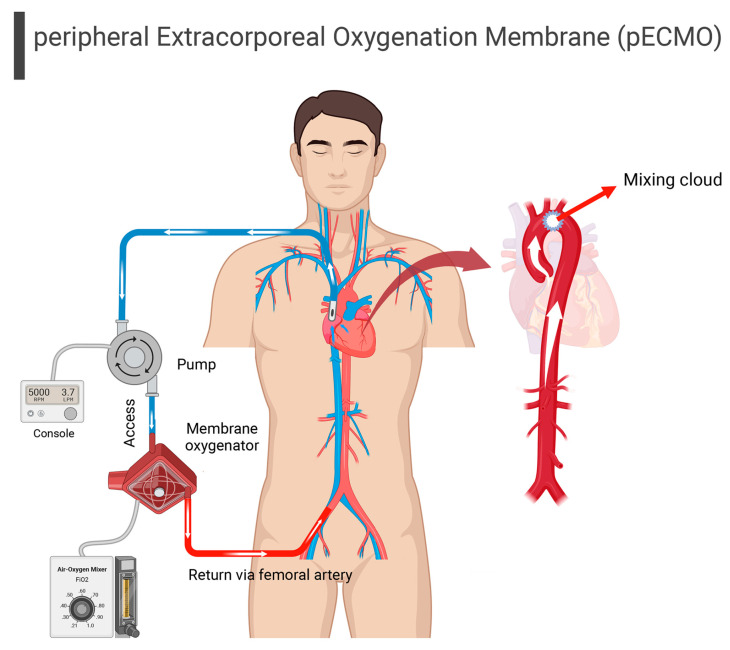
Peripheral extracorporeal membrane oxygenation (p ECMO). Blood returns to the body via the femoral artery with unnatural blood flow dynamics. In the presence of heart function, the two circulations meet at a point known as the mixing cloud; the location of the mixing cloud determines whether the organs are perfused by natural or ECMO blood flow.

**Table 1 brainsci-13-01165-t001:** Characteristic of each pattern of ECMO-related ABIs.

Patterns Of ABI	Incidence	Intracranial Location	Timing
ICH	VV: 2–18%, VA: 4–19%	frontal neocortex, basal ganglia, cerebellum, pons	shortly after cannulation to 7 days cannulation
Ischemic Stroke	VV: 1.7–2% VA: 3.6–5.3%	frontal neocortex, basal ganglia anterior hypophysis cerebellum	second week of ECMO support
Seizures	VA: 2–6% VV: 1.3%	MCA territory	3.2 h of the initiation of cEEG monitoring
HIBI	VV: 1% VA: 13%	cerebral cortex, cerebellum brain stem, basal ganglia	peri-ECMO cannulation period
Brain Death	VV: 2% VA: 7.9%	NA	NA

ECMO: extracorporeal membrane oxygenation; ABI: acute brain injury; VV: venovenous; VA: venoarterial; ICH: intracranial hemorrhage; HIBI: hypoxic–ischemic brain injury; cEEG: continuous electroencephalogram; MCA: middle cerebral artery.

## Data Availability

Data sharing not applicable. No new data were created or analyzed in this study. Data sharing is not applicable to this article.

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
