# Peer review of "Subtypes and Mechanistic Advances of Extracorporeal Membrane Oxygenation-Related Acute Brain Injury"

_brainsci, 2023, doi:10.3390/brainsci13081165_

Round 1

Reviewer 1 Report

Dear Authors,

Please check the font of the manuscript

This paper is fascinating and innovative: let me suggest expanding on some points, such as

Role of Thromboelastography, ROTEM, and ACT to monitor possible bleeding and thrombotic complications

Please consider the use of Filter or hemoadsorption in ECMO

Reviewer 2 Report

Thank you for the opportunity to preview your review of brain injury mechanisms in ECMO. Overall I found it to cover a wide range of potential mechanisms for brain injury and appreciate the large amount of work it is to focus such a broad topic. I have a few recommendations to consider to, thank you for your time.          

Line 10 – Abstract; The statement that ECMO is a highly effective mechanical circulatory support technique is controversial, would consider rephrasing to something such as frequency of use.

Line 31 – Introduction; Would consider adding hypercarbia or ventilation failure as a potential indication or rephrase.

Line 35 – Introduction; Would rephrase how you describe central ECMO sites, similarly consider your definition for peripheral as it is usually but not always within the femoral vessels.

Line 40 – Introduction; Your first statement is controversial, would consider changing efficacy to potential and add citations to support.

Line 95 – Ischemic Stroke; Just a general comment, this paragraph presents evidence that ischemic stroke is more common in VA ECMO, but the potential mechanisms pertain to VV ECMO patients (examples: venous sinus thrombosis with a PFO, internal jugular cannulation, and dual lumen catheters).

Line 133 – Hypoxic-Ischemic Brain Injury; General comment you mention respiratory failure as a potential etiology for HIBI being higher in VA ECMO (perhaps more associated with VV ECMO?), any consideration to mention or consider differential oxygenation/North South syndrome.

Line 309 – Cannulation method; Can you specify if you are discussing adults or children/infants?

Line 407 – Conclusion; would reword or omit your first sentence as you didn’t focus on the connection of ABI to mortality.

One general statement – overall it is sometimes unclear if you are referring to infants, children, or adults; and VV or VA ECMO. I may be helpful to label or organize them to better understand which population you are referring to if it is a certain cohort or all patients.

Overall quality of English language is very good/fluent maybe minor revisions could be made.

Reviewer 3 Report

With a great interest I read a work of Deng et al. on the Subtypes and Mechanistic Advances of Extracorporeal Membrane Oxygenation Related Acute Brain Injury.

Authors should be congratulated for a great subject selection, and well done work.

However, I have few recommendations to further improve this work:

Line 11: please remove word “technique”.

Line 12-13: please change the order of complications based on the realistic incidence “However, ECMO is associated with several complications, such as acute brain injury (ABI), thromboembolism, and acute kidney injury” into “However, ECMO is associated with several complications, such as acute kidney injury, haemorrhage, thromboembolism, and acute brain injury (ABI)”.

Line 13 and 14: please remove plural form from haemorrhage and infarction.

Line 29: please replace the word “removed” with the word “drained”.

Line 40-41: reformulate into “Despite the potential benefits of ECMO, this life-saving technology may be associated with serios complications, including acute brain injury (ABI).”

Line 41: we dont start a sentence with shortcut usually.

Line 43-46: please reformulate into “Studies based on the Extracorporeal Life Support Organization Registry (ELSO) have reported ABI in 11–20% of adult patients undergoing VA-ECMO, and the most recent meta-analysis excluding data from ELSO up to 12.5%”. Here please use the citation: https://pubmed.ncbi.nlm.nih.gov/36195759/

Line 64-66 is repetition of statement from introduction. Please delete or reformulate to avoid repetition.

Line 89-93: please order the reasons based on the logical importance. PaCO2 is rather less important compared to anticoagulation. High transfusion requirement should be removed, it is rather a consequence of bleeding.

Section 2.2 ischemic stroke: please cite the most recent work on thrombosis / stroke during ECMO (https://pubmed.ncbi.nlm.nih.gov/37176673/)

Line 147-148: please cite the information from the most recent meta-analysis on brain death in VA ECMO: https://pubmed.ncbi.nlm.nih.gov/36195759/

Lines 181-190: please add a sentence on still missing the most appropriate anticoagulation monitoring tests, and that the levels of aPTT did not show any association with bleeding or thrombosis (reference: https://pubmed.ncbi.nlm.nih.gov/37244207/ and https://pubmed.ncbi.nlm.nih.gov/37176673/)

Line 204-206: please reformulate into similar to “However, inadequate systemic anticoagulation increases the risk of bleeding in case of overdosing, or thrombosis in case of underdosing. Therefore, it is essential to monitor anticoagulation and balance procoagulant and anticoagulant factors [68,69].”

Congratulation on great figures.

Change the caption of the title 3.2 into (or similar) Mechanisms of ABI specific to VA-ECMO

Line 289: we should not compare the two types of ECMO on incidence, as they are indicated in different diseases. Please remove the sentence “Among the two modes of ECMO, VA-ECMO is more commonly used in patients”

Please adapt the title of section 3.3 into (or similar) Mechanisms of ABI specific to VV-ECMO

Please correct the heading numbering in line 367 and on.

Line 368: “Patients who need ECMO support are in critical illness” patients are not “in” but suffering from critical illness or state simply “critically ill patients receiving ECMO support...”.

Line 368-369: please reorder this into: cardiac shock, cardiac arrest, or respiratory failure. ARDS would be within the respiratory failure?

Please add a section: Future perspectives and development, where you may describe the possible methods of prevention of ABI and reduction of their incidence.  

English language and flow need to be improved. 
